# Immunophenotypic Alterations in Adult Patients with Steroid-Dependent and Frequently Relapsing Nephrotic Syndrome

**DOI:** 10.3390/ijms24097687

**Published:** 2023-04-22

**Authors:** Federica Casiraghi, Marta Todeschini, Manuel Alfredo Podestà, Marilena Mister, Barbara Ruggiero, Matias Trillini, Camillo Carrara, Olimpia Diadei, Alessandro Villa, Ariela Benigni, Giuseppe Remuzzi

**Affiliations:** 1Istituto di Ricerche Farmacologiche Mario Negri IRCCS, 24126 Bergamo, Italy; 2Unit of Nephrology and Dialysis, Azienda Socio-Sanitaria Territoriale Papa Giovanni XXIII, 24127 Bergamo, Italy

**Keywords:** nephrotic syndrome, steroid-dependent nephrotic syndrome, steroid-resistant nephrotic syndrome, B cells, regulatory T cells, rituximab

## Abstract

Immune dysregulation plays a key role in the pathogenesis of steroid-dependent/frequently relapsing nephrotic syndrome (SDNS/FRNS). However, in contrast with evidence from the pediatric series, no major B- or T-cell alterations have been described for adults. In these patients, treatment with rituximab allows safe discontinuation of steroids, but long-term efficacy is variable, and some patients experience NS relapses after B cell reconstitution. In this study, we aimed to determine disease-associated changes in the B and T cell phenotype of adult patients with SDND/FRNS after steroid-induced remission. We also investigated whether any of these changes in immune cell subsets could discriminate between patients who developed NS relapses after steroid-sparing treatment with rituximab from those who did not. Lymphocyte subsets in SDNS/FRNS patients (*n* = 18) were compared to those from patients with steroid-resistant NS (SRNS, *n* = 7) and healthy volunteers (HV, *n* = 15). Before rituximab, SDND/FRNS patients showed increased frequencies of total and memory B cells, mainly with a CD38-negative phenotype. Within the T-cell compartment, significantly lower levels of FOXP3^+^ regulatory T cells (Tregs) were found, mostly due to a reduction in CD45RO^+^ memory Tregs compared to both SRNS and HV. The levels of CD45RO^+^ Tregs were significantly lower at baseline in patients who relapsed after rituximab (*n* = 9) compared to patients who did not (*n* = 9). In conclusion, patients with SDND/FRNS displayed expansion of memory B cells and reduced memory Tregs. Treg levels at baseline may help identify patients who will achieve sustained remission following rituximab infusion from those who will experience NS relapses.

## 1. Introduction

Nephrotic syndrome (NS) is a clinical condition that is common to a heterogeneous group of glomerular diseases and is characterized by diffuse edema, heavy proteinuria and hypoalbuminemia.

In children and young adults, most cases of NS are associated with alterations of podocyte structure and function, which lead to pathological modifications of the glomerular filtration barrier. Histological patterns in most cases are consistent with minimal change disease (MCD) and, albeit less frequently, focal segmental glomerulosclerosis (FSGS) [1,2,3].

The prognosis of these disorders is dependent on response to steroids, with up to 80% of cases achieving NS remission after a course of oral prednisone. However, after the initial response, about 70% of adult patients experience frequent NS relapses that require further treatment or develop steroid dependence [4,5,6]. According to the timing and rate of relapses, these patients are classified as steroid-dependent (SDNS) or frequently relapsing (FRNS) nephrotic syndrome [7]. Both conditions are associated with increased cumulative exposure to glucocorticoids with deleterious long-term side effects [6]. To avoid chronic steroid therapy in SDNS/FRNS patients, a number steroid-sparing agents have been proposed. Among them, the anti-CD20 monoclonal antibodies rituximab and ofatumumab have been shown to induce sustained remission, allowing safe steroid tapering in the majority of patients [8,9,10].

The sensitivity of NS to steroid therapy and immunosuppressive drugs and the efficacy of B cell depletion in inducing remission of the NS strongly suggests a key role of a systemic dysregulation of the immune system in the pathogenesis of steroid-sensitive forms of the disease [11,12]. The intense investigation, mainly in pediatric patients with NS, reported T cell alteration characterized by reduced CD4^+^ T cell levels and increased cytotoxic CD8^+^ T cells [13,14,15], higher effector Th2 [16] and Th17 [17,18] cells and impaired regulatory T cell (Treg) frequency or activity [19,20]. More recently, steroid-sensitive forms of pediatric NS have been associated with increased B cell levels [21] at disease onset and relapse with the main role of B cells with a switched-memory phenotype [22]. Studies analyzing B cell repopulation post-rituximab reported that the emergence of this switched-memory B cell subset predicted the recurrence of NS in pediatric patients [23,24], providing a possible explanation of why some patients experience disease relapse after B cell recovery, whereas other patients maintain a long-term remission even after complete B cell recovery.

However, the described T and B cell alterations were not consistently reported in other studies, and no major immune dysregulation has been described for adult patients with NS [11,12,21]. These conflicting findings may be due to the complexity of disease mechanisms and could suggest that different immunological alterations could occur in specific forms of the disease, therefore explaining the diverse response to treatment.

In this study, we characterized the B and T cell phenotype in a selected cohort of adult patients with SDND/FRNS during a remission phase induced by steroid treatment and before therapy with rituximab as a steroid-sparing agent. We compared lymphocyte subsets in these patients with those from SRNS patients under similar steroid therapy and those from healthy subjects to assess disease-specific phenotypes. In addition, we aimed to evaluate whether baseline changes in B and T cell subsets could identify SDND/FRNS patients who relapse or do not relapse after rituximab infusion.

## 2. Results

### 2.1. Baseline Patient Characteristics

Eighteen adult patients with SDNS/FRNS who were treated with rituximab from February 2010 to October 2017 were included in the study. Before rituximab infusion, all SDNS/FRNS patients had achieved partial (16.7%) or complete (83.3%) remission of the nephrotic syndrome with prednisone, and two patients were also receiving either cyclosporin or mycophenolate mofetil as steroid-sparing agents. We also included seven patients with SRNS and 15 age- and sex-matched healthy volunteers (HV) as controls in all the analyses.

Consistent with NS remission, patients with SDNS/FRNS showed significantly lower levels of serum creatinine and proteinuria and higher levels of serum albumin compared to SRNS patients (Table 1). Most SDNS/FRNS patients were males (55%) with a median age of 40.5 (range: 27–47 years) 7, and minimal change disease was the most frequent histological pattern. Less than half of SDNS/FRNS patients had a pediatric onset of the disease, and steroid dependence was more common than the frequently relapsing clinical presentation (Table 2). The average dose of prednisone in the last month before rituximab infusion was higher in the SDNS/FRNS group compared to that of SRNS patients, but the difference did not reach statistical significance. The non-negligible dose of steroids assumed by SRNS patients was largely due to the timing of sample collection: most patients were sampled shortly after the diagnosis of steroid resistance had been made, and they were not completely weaned off corticosteroids at that time.

After rituximab infusion, immunosuppressive drugs were progressively weaned over time until discontinuation which was achieved on average after 4.8 ± 1.4 months. During the two year follow-up period, nine (50%) of the SDNS/FRNS patients experienced at least one NS relapse, with a mean time to the event of 9.9 ± 3.9 months. Baseline characteristics of patients with or without NS relapses were similar, including the time required to completely wean immunosuppression (5.1 ± 1.3 months and 4.3 ± 1.4 months in relapsers and non-relapsers, respectively). Notably, baseline serum albumin was significantly higher in non-relapsers compared to relapsers (Table 2).

A scheme of the subject groups and of the experimental design is provided in Figure 1.

Patients with SDNS/FRNS displayed significantly increased percentages of CD19^+^CD20^+^ B cells among total circulating lymphocytes compared to both SRNS patients and HV (Figure 2A). Compared to HV, the increase in B cells in SDNS/FRNS patients was mainly due to increased memory B cells as documented by significantly higher percentages of CD38^-/low^CD24^high^ memory B cells (Figure 2B,C), total memory CD27^+^ B cells, as well as both IgM memory and switched memory B cells (Figure 2D,E). The levels of mature B cells in SDNS/FRNS patients were similar to those of HV, whereas transitional B cells in this group were significantly lower compared to healthy volunteers (Figure 2B,C).

SRNS patients shared with the SDNS/FRNS group higher-than-normal percentages of switched memory B cells (Figure 2E) and significantly lower percentages of transitional B cells (Figure 2C).

Higher percentages of CD27^+^ B cells on total circulating lymphocytes in SDNS/FRNS patients than in HV and SRNS patients, mainly of unswitched phenotype, were confirmed by B cell classification by IgD/CD27 expression (Appendix A). Similarly, significantly higher levels of CD38^-/low^CD24^high^ memory B cells and lower levels of transitional and mature B cells in SDNS/FRNS compared to HV were confirmed also when the different B cell subsets were expressed as percentages of total B cells (Figure 3A). Moreover, total CD27^+^, IgM memory and switched memory B cells were calculated as percentages of B cells were similar in SDNS/FRNS patients and HV (Figure 3B).

Lower CD38^+^ B cell levels in SDNS/FRNS were confirmed by the significantly lower percentages of Bm3+4, Bm2′ and Bm2 of the IgD/CD38 B cell subsets calculated as either percentages of total B cells or of lymphocytes (Appendix A).

The shift in CD38/CD24 B cell phenotype toward higher than normal levels of CD38^-/low^ memory B cells and very low levels of transitional B cells was also observed in SRNS patients (Figure 3A).

To assess whether the decreased frequency of transitional B cells and increased memory/mature B cells within the B cell compartment could be due to the concomitant steroid therapy in FRNS/SDNS and SRNS patients, we explored the correlation between the frequencies of B cell subsets and PDN dose, pooling data from the two groups of NS patients (Figure 3C). The percentages of transitional B cells inversely correlated with PDN dose, whereas percentages of mature and memory B cells within the CD24/CD38 B cell compartment did not. We also performed correlation analyses between PDN dose and all the other B cell subpopulations and did not find any significant correlation, indicating that, in our cohort, transitional B cells are the only B cell subpopulation significantly affected by the oral steroid treatment.

SRNS patients displayed a significant increase in total memory CD27^+^ B cell percentages compared to HV with switched memory B cells accounting for most of the difference. Notably, switched memory B cells were significantly higher in SRNS patients compared to the SDNS/FRNS group.

Characterization of memory CD27^+^ B cells (IgD/CD27 B cell classification) in SRNS revealed a significant increase in switched and double negative B cell frequencies at the expense of reduced unswitched and naïve B cells, compared to both HV and SDNS/FRNS patients (Appendix A).

Overall, these findings document an increase in memory B cells in both SDNS/FRNS and SRNS patients that are characterized by a CD38-negative phenotype in SDNS and by an IgD-negative phenotype in SRNS. Both patient groups displayed markedly low levels of transitional B cells.

Percentages of CD8^+^ and CD4^+^ T cells among circulating CD45^+^ lymphocytes in SDNS and SRNS were comparable to those from HV (Figure 4A). Consistently, we did not observe any significant difference in the frequency of central memory, effector memory and naïve CD8 and CD4 T cell subsets among these groups (Appendix A), with the exception of a significantly lower frequency of central memory CD8^+^ T cells in both SDND/FRNS and SRNS compared to HV.

When we considered regulatory CD4^+^ T cell subsets, SDNS/FRNS patients displayed a lower frequency of Tregs, mostly due to a reduction in those with a memory phenotype marked by CD45RO^+^ expression (Figure 4C). Treg percentages in SRNS patients were comparable to HV, except for a significant increase in naïve Tregs (CD45RA^+^) compared to both HV and SDND/FRNS (Figure 4C). Overall, SDNS/FRNS patients appeared to have lower Treg frequency, mainly of the memory phenotype, a defect that was not observed in SRNS patients.

### 2.2. Memory Treg Are Lower in SDNS/FRNS Patients Who Relapse after Rituximab

In-depth characterization of the B cell phenotype before rituximab administration did not reveal any significant difference between SDNS/FRNS patients who relapsed and those who did not (Appendix A). In addition, no differences in the percentage of naïve and memory CD8 and CD4 T cell subsets were found between these groups (Appendix A), even when naïve and memory CD8 and CD4 T cell subpopulations were expressed as counts per microliter (Appendix A). The frequency (Figure 4A) and counts (Appendix A) of FOXP3^+^ total Tregs were lower in relapsers compared to non-relapsers, even though the difference did not reach statistical significance. Although CD45RA^+^ naïve Treg levels (Figure 5A and Appendix A) were similar between the two patient groups, we observed at the baseline a significantly lower level of CD45RO^+^ memory Treg percentages (Figure 5A) and counts (Appendix A) in SDNS/FRNS patients who relapsed compared to those who did not. These changes were independent of the dose of steroids that these patients were receiving (Figure 5B).

We also performed a subgroup analysis comparing B and T cell subsets between SDND/FRNS patients with histological diagnosis of MCD (n = 11) or FSGS (n = 7). We did not find any significant difference between B and T cell subpopulations between the two patient groups (Appendix A). However, we identified a trend towards higher FOXP3^+^ total and CD45RO^+^ Tregs and lower CD8^+^TEMRA in MCD compared to FSGS patients (Appendix A). The small sample size precluded us from drawing any definitive conclusion on the possible differences in B and T cell phenotypes between these two SDNS/FRNS patient groups.

## 3. Discussion

Sensitivity to steroids and to immunosuppressive drugs strongly supports the involvement of B and T cell alterations in the pathogenesis of NS. However, the response to steroids is variable with subsets of patients developing steroid dependence while others are steroid resistant. For patients with SDNS/FRNS, B cell depletion with rituximab is an effective therapy, allowing tapering and withdrawal of steroid treatment. Moreover, in this setting, some patients experience long-term remission off therapy while others may develop NS relapses after B cell recovery [10,23]. This may suggest that a different immune cell dysregulation characterizes SDNS/FRNS and altered B or T cell subsets can discriminate between patients who relapse post-rituximab and those who do not.

Before rituximab infusion, our cohort of adult SDNS/FRNS patients in the steroid-induced remission phase was characterized by increased frequencies of B cells due to an increase in memory B cells and a marked reduction in transitional B cells. This finding has been reported by several studies—conducted mainly in steroid-sensitive pediatric patients—documenting increased levels of B cells at disease onset [22,25,26], even though some other reports found opposite results [13,27]. In-depth phenotypic evaluation of expanded B cells in children with SSNS showed that the increase in the frequency of total B cells at the onset of disease was primarily due to transitional [22,26], memory [22,26] and, in particular, switched memory B cell [22,26] expansion. At remission of the disease, these B cell subsets were comparable to healthy controls whereas, in relapsing patients, total memory [22,26] including switched memory B cells [22] were significantly higher, despite the immunosuppressive treatment.

In contrast with the above findings, despite being in steroid-induced remission, our cohort of adult FRNS/SDNS patients showed higher than normal frequencies of memory B cell subsets, which were significantly higher also when compared to patients with SRNS. In addition, we observed that the increase in memory B cells was associated with very low levels of transitional B cells. Lower than normal levels of transitional B cells were already reported in SSNS pediatric patients during steroid treatment [28] and in FRNS/SDNS children in steroid-induced complete remission phase [23], suggesting that low levels of transitional B cells may be the result of concomitant immunosuppression. According to these studies, our findings that the percentages of transitional B cells within the B cell compartment were reduced either in FRNS/SDNS or SRNS patients and inversely correlated with the prednisone dose in this whole cohort strongly support the evidence that steroids induced a shift toward reduced transitional B cells in adult NS patients, similarly to pediatric patients. However, the evidence that total B cells as well as memory CD38^-^ B cells and memory CD27^+^ B cell frequencies were selectively increased in FRNS/SDNS compared to both healthy volunteers and SRNS patients under steroid treatment argues against an impact of steroids on memory B cell changes, suggesting, instead, a role of memory B cells in the pathogenesis of adult FRNS/SDNS.

The only available study analyzing B cell dysregulation in adult patients with NS has been conducted in patients with biopsy-proven minimal change disease [27]. In this study, B cell frequency and percentages of transitional, naive and memory B cell subsets were comparable in patients with NS in the active phase (at first episodes or at relapse) and in patients in remission and in healthy volunteers. The only population of B cells expanded in MCD patients with active NS were plasmablasts [27], the population, however, was found to be at normal levels in SSNS children [29]. The discrepancy in the B cell results in adult NS patients could be explained by the different populations of patients considered in our study and in the report from Oniszczuk and colleagues. In our study, we considered only patients with FRNS/SDNS with either MCD or FSGS, whereas Oniszczuk et al. studied patients with MCD that likely include most steroid-sensitive subjects. Patients with FRNS/SDNS could bear specific B cell dysregulations not evident in steroid-sensitive patients, thus justifying the different results obtained in our cohort.

We did not find any difference in B cell subsets able to distinguish at baseline FRNS/SDNS patients who did or did not relapse after rituximab therapy. A similar finding was already reported in a pediatric cohort, where no changes in B cells were found at baseline in FRNS/SDNS patients before rituximab therapy. In this study, analysis of B cell repopulation following B cell depletion induced by rituximab showed that total, IgM and switched memory B cells recovered earlier in relapsers than in non-relapsers, and the recovery of switched memory B cells was strongly predictive of relapse in children [23]. We did not evaluate the phenotype of B cells during reconstitution in our cohort of patients, and, therefore, we cannot exclude that a similar profile of memory B cell recovery might predict NS relapse also in adult patients.

Several alterations in T cell subpopulations have been described in NS. A decrease in Foxp3+ regulatory T cells has been consistently found in children [18,30] as well as in adult patients at the onset of NS [17]. Consistent with a possible role in disease pathogenesis, Treg levels returned to normal levels during remission induced by steroids [30,31] or by rituximab [19,32].

The extent of the increase in circulating Treg levels from onset to steroid-induced remission in SSNS pediatric patients was lower in patients who subsequently developed frequent relapses compared to those who achieved sustained remission [33]. Moreover, the development of MCD has been observed in a patient with IPEX syndrome, a rare condition with defective Tregs caused by a genetic mutation in the *foxp3* gene, further implicating Tregs in the disease [34].

In agreement with these studies, we found reduced Treg levels, in particular memory Tregs, in patients with FRNS/SDNS compared to healthy volunteers and to SRNS patients. At baseline, patients who relapsed following rituximab therapy showed significantly lower level of memory Tregs compared to those experience sustained remission. This finding confirms the results of a previous study in a pediatric cohort of patients with SDND and SRNS and biopsy-proven MCD on CNI-based immunosuppression [35]. In this study, lower Treg levels at baseline predicted the occurrence of early NS relapse post-rituximab with high sensitivity and specificity [35].

In our patients, the total population of Tregs was only numerically higher in non-relapsing than in relapsing patients, whereas a significant difference was reached with memory Tregs. This could be explained by differences in the Treg pool between children and adults as supported by the evidence that the proportion of naïve Tregs declines with age and the memory Treg increases [36].

The mechanism linking low levels of Tregs with disease recurrence post-rituximab remains elusive. Experimental evidence supports a direct association between a defect in Tregs and proteinuria development during a trigger event, such as adriamycin-induced nephropathy [37], in an animal model of FSGS, Buffalo/Mna rats [38] and LPS nephropathy [39]. On the other hand, Tregs can directly affect B cells, inhibiting both antibody production [40] and class-switch recombination [41]. Conversely, B cells can inhibit Treg generation by preferentially expanding activated and effector T cells [42]. Consistently, Tregs increase after B cell depletion with rituximab [42]. Therefore, the involvement of reduced Tregs in NS pathogenesis does not necessarily exclude a complementary pathogenic role of B cells, a complex dysregulation that could be reversed by rituximab when Tregs are present in a sufficient number. Unfortunately, we did not evaluate Treg and B cell changes in the relapsing phase after B cell depletion and reconstitution which would have been critical in investigating T and B cell interaction.

Although it was not the focus of our work, we observed that patients with SRNS showed increased percentages of memory and switched memory B cells compared to either healthy control and patients with FRNS/SDNR and normal Treg levels. These results are in contrast to previous studies showing normal levels of memory B cell subsets [26] and reduced levels of Tregs [16,31], including memory Tregs [16] in children with SRNS. These contrasting findings would suggest that adult SRNS could be mediated by different pathogenic immune mechanisms compared to pediatric SRNS, highlighting the well-known complexity and heterogeneity of this condition. Whether the observed alteration in B and T cells actually plays a pathogenic role in adult SRNS merits further investigation.

The limitations of this study are the small sample size and the retrospective analysis which are direct consequences of the rarity of the disease in adult patients.

In conclusion, our findings may be useful for the improvement of therapeutic approaches in adult patients with FRNS/SDNS. If confirmed, Treg levels at baseline may help identify those patients who will achieve sustained remission following rituximab therapy from those for whom alternative strategies should be considered.

## 4. Materials and Methods

### 4.1. Patients and Definitions

In this retrospective cohort study, we enrolled adult patients with biopsy-proven MCD or FSGS with a history of frequent relapses or steroid dependence who achieved complete or partial NS remission with steroids and were candidates to receive rituximab as a steroid-sparing agent. Steroid dependence was defined by at least one NS relapse during steroid therapy or within 15 days from its discontinuation, whereas the occurrence of two or more relapses off steroid therapy within any six month period led to the diagnosis of frequently relapsing NS.

Complete and partial NS remission were defined, respectively, by urinary protein excretion < 0.3 g/day along with normal serum albumin concentration and stable renal function, and urinary protein excretion < 3.5 g/day or with ≥50% reduction from baseline values, accompanied by an improvement or normalization of serum albumin concentration and stable renal function. NS relapses were defined by urinary protein excretion > 3.5 g/day after remission had been obtained. Only SDNS/FRNS patients with at least two years of follow up were included in the study, and patients were classified as relapsers or non-relapsers based on the occurrence of at least one NS relapse within this timeframe.

Patients with steroid-resistant MCD or FSGS (defined as persistent NS despite therapy with prednisone 1 mg/kg/day for at least four months) with no pathogenic mutations for known genes associated with NS, as well as age- and sex-matched healthy volunteers were also enrolled in the study as controls. Subjects with a reasonable possibility of a secondary cause of NS, including active systemic infections, malignancy or drug abuse were excluded from the study.

Patients were identified among the subjects treated at the Nephrology Unit of the Azienda Socio-Sanitaria Territoriale Papa Giovanni XXIII and followed up at the Clinical Research Centre for Rare Diseases “Aldo e Cele Daccò” of the Istituto di Ricerche Farmacologiche Mario Negri IRCCS (Bergamo, Italy). Anonymized patient-level data were retrospectively collected from medical charts up to the last available visit and documented on site into dedicated electronic Case Report Forms. The study was approved by the Ethics Committee of Azienda Socio-Sanitaria Territoriale Papa Giovanni XXIII (12 October 2018), and all participants provided written informed consent in compliance with the Declaration of Helsinki.

### 4.2. Procedures and Outcomes

As per center protocol, rituximab was given as a single dose (375 mg/m^2^), and additional infusions were planned only if complete B cell depletion (i.e., CD19^+^ B cell counts <5 cells/µL) was not achieved after one week from the first one. Concomitant immunosuppression was progressively tapered over 6–9 months starting from one month after rituximab therapy up to complete withdrawal or NS recurrence. Steroid tapering was started after complete weaning from other immunosuppressive drugs.

Blood samples were collected before treatment with rituximab in SDNS/FRNS patients, and peripheral blood mononuclear cells (PBMC) were isolated by Ficoll-Paque Plus density gradient centrifugation. PBMC was suspended in RMPI 20% AB serum 10% DMSO and stored in liquid nitrogen until the day of the analysis.

The primary outcome of the study was the difference in the frequency of B- and T-cell subsets among patients with SDNS/FRNS, SRNS and healthy volunteers. As a secondary outcome, we evaluated differences in B- and T-cell subsets between SDNS/FRNS who experienced a relapse within two years from rituximab infusion and those who did not.

### 4.3. Flow Cytometry Analysis

PBMCs were stained with fluorochrome-conjugated murine monoclonal antibodies against human CD3 (clone HIT3a), CD4 (clone RPA-T4), CD8 (clone RPA-T8), CD45RA (clone HIT100), CD45RO (clone UCHL1), CD25 (clone M-A251), CD127 (clone HIL-7R-M21), CD19 (clone HIB19), CD27 (clone M-T271), CD38 (clone HIT2), IgD (clone IA6-2), CD24 (clone ML5) and FoxP3 (clone 236A/E7) (all from BD Biosciences, San Jose, CA, USA). Multicolor flow cytometry was used to identify T- and B-cell subsets with standard technique and equipment (FACS FortessaX20, BD Biosciences and Flow-jo software, version 10.7.1).

### 4.4. Statistical Analysis

Baseline characteristics were analyzed through descriptive statistics, and presented as number (%), mean ± SD or median [IQR] as appropriate. Comparisons between two groups were analyzed with a *t*-test or Mann–Whitney test depending on the shape of the distribution. One-way ANOVA or Kruskal–Wallis tests were used to assess differences among more than two groups, and follow-up tests were performed in case of significant differences. Significance was inferred at the 5% probability level, and all *p*-values were two sided. Data were analyzed with GraphPad Prism (version 9.3.1).

## Figures and Tables

**Figure 1 ijms-24-07687-f001:**
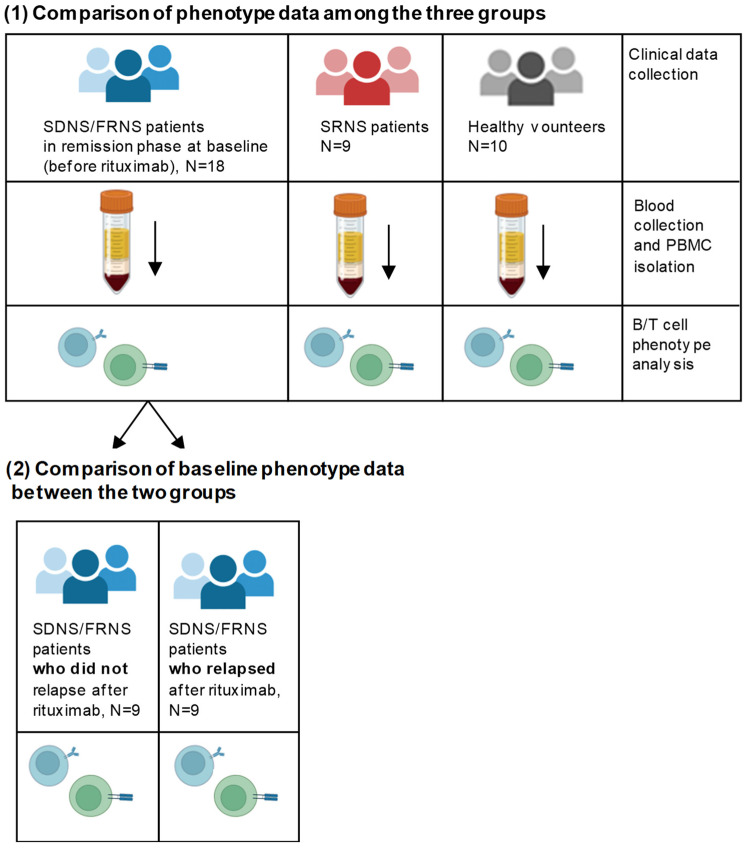
Subject groups and scheme of the experimental study design.

**Figure 2 ijms-24-07687-f002:**
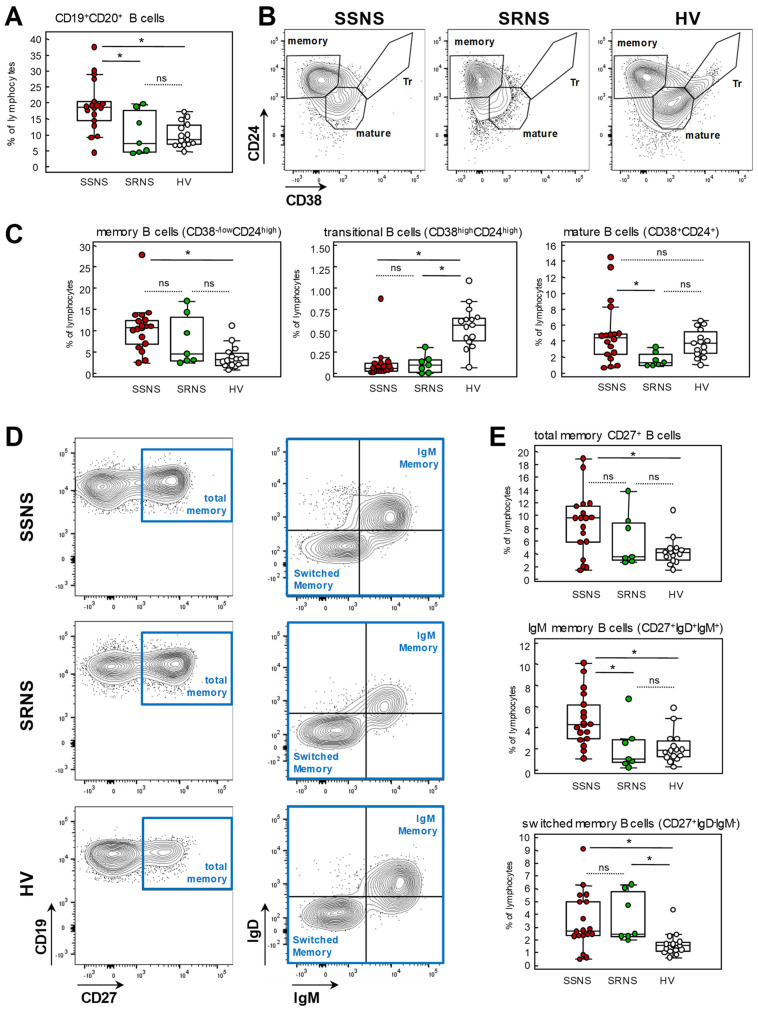
FACS analysis of circulating B cell subsets in SSNS, SRNS patients and HV. (**A**) Percentages of CD19^+^CD20^+^ B cells among total lymphocytes in the three groups of subjects; (**B**) Representative contour plots of memory, mature and transitional B cell gating strategy in SSNS, SRNS patients and HV; (**C**) Percentages of memory, transitional and mature B cells among total lymphocytes in patients with SSNS or SRNS and in HV; (**D**) Representative contour plots of IgM memory and Switched memory on gated CD27^+^ memory B cells and (**E**) the relative percentages among total lymphocytes in the three groups of subjects. Plots display the median, 25th and 75th percentiles of distribution (boxes) and whiskers extend to the minimum and maximum values of the series. * *p* < 0.05 between the indicated groups; ns: not significant.

**Figure 3 ijms-24-07687-f003:**
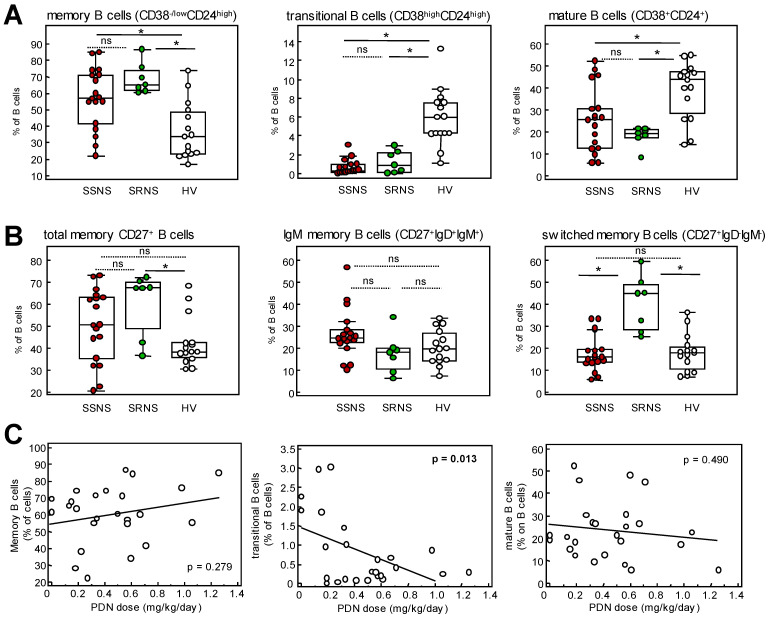
FACS analysis of memory B cell subsets and correlation with prednisone dose. Percentages of (**A**) memory, transitional and mature B cells and (**B**) total, IgM and switched memory B cells on CD19^+^CD20^+^ B cells in SSNS and SRNS patients and HV. Plots display the median, 25th and 75th percentiles of distribution (boxes) and whiskers extend to the minimum and maximum values of the series; * *p* < 0.05 between the indicated groups; ns: not significant. (**C**) Correlation between percentages of memory, transitional and mature B cells with prednisone dose in NS patients. Correlations were analyzed using Pearson’s correlation coefficient.

**Figure 4 ijms-24-07687-f004:**
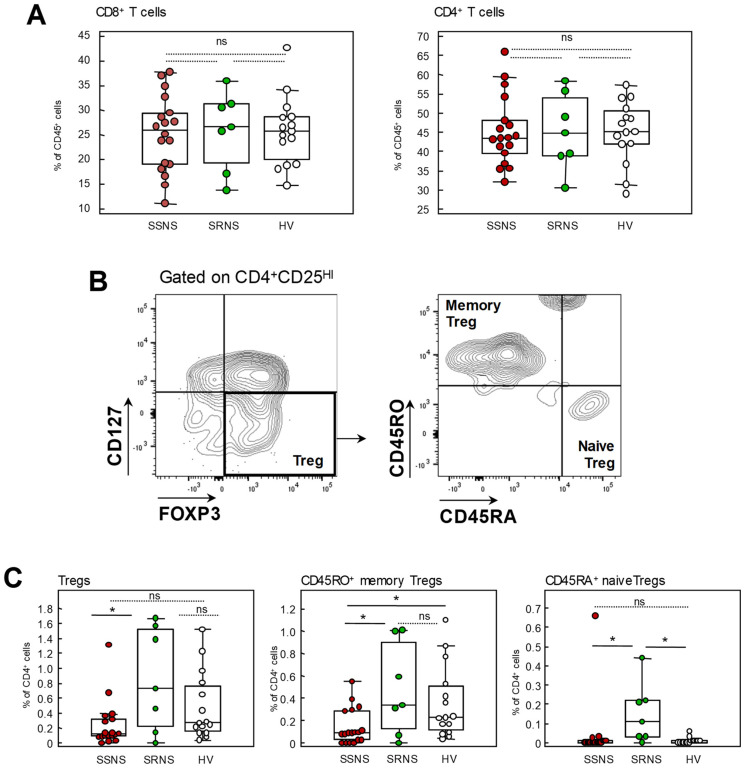
Regulatory T cells are reduced in SDNS/FRNS patients. (**A**) Percentages of CD3^+^CD8^+^ and CD3^+^CD4^+^ T cells on circulating CD45^+^ cells in SSNS, SRNS patients and HV; (**B**) gating strategy for the identification of memory (CD45RO^+^) and naïve (CD45RA^+^) Tregs on FOXP3^+^CD127- Tregs gated on CD4^+^CD25hi cells; (**C**) percentages of total, memory and naïve Tregs on CD4^+^ T cells in the three groups of subjects. Plots display the median, 25th and 75th percentiles of distribution (boxes) and whiskers extend to the minimum and maximum values of the series; * *p* < 0.05 between the indicated groups, ns: not significant.

**Figure 5 ijms-24-07687-f005:**
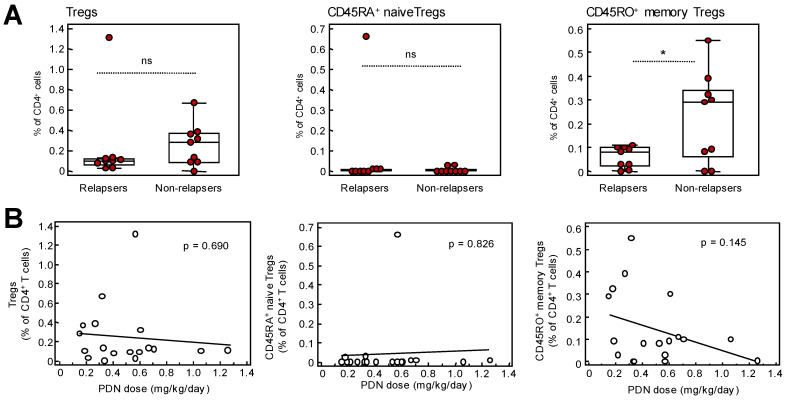
Treg levels in relapsers and non-relapsers SDNS patients. (**A**) Percentages of total, memory and naïve Tregs on CD4^+^ T cells in SDNS/FRNS patients who relapsed or did not after rituxi-mab therapy and (**B**) their correlation with prednisone dose. Plots display the median, 25th and 75th percentiles of distribution (boxes) and whiskers extend to the minimum and maximum values of the series, correlations were analyzed using Pearson’s correlation coefficient. * *p* < 0.05 between the indicated groups, ns: not significant.

**Table 1 ijms-24-07687-t001:** Baseline characteristics of patients with steroid-dependent/frequently relapsing nephrotic syndrome (SDNS/FRNS, with steroid-resistant nephrotic syndrome (SRNS) and healthy volunteers (HV). FSGS: focal segmental glomerulosclerosis; MCD: minimal change disease; PDN: prednisone equivalents. Data are expressed as median [interquartile range] or number (percentage). *p*-values refer to the comparison between SDNS/FRNS and SRNS patients.

	SDNS/FRNS (*n* = 18)	SRNS (*n* = 7)	HV (*n* = 15)	*p*-Value
Age (years)	40.5 [27–47]	31 [24–50]	44 [37–54]	0.286
Male sex, n (%)	10 (55%)	2 (29%)	7 (47%)	0.4775
MCD vs. FSGS	11 vs. 7	2 vs. 5	-	0.2016
Pediatric onset, n (%)	8 (44%)	0 (0%)	-	0.0573
Serum creatinine (mg/dL)	0.80 [0.75–0.94]	1.19 [0.82–2.05]	-	0.0458
Serum albumin (mg/dL)	3.7 [3.6–4.0]	1.75 [1.38–2.03]	-	0.0008
Urinary protein (g/24 h)	0.09 [0.07–0.17]	14.28 [5.26–18.83]	-	0.0001
Average PDN dose (mg/kg/day)	0.47 [0.270–0.610]	0.19 [0.032–0.545]	-	0.1228

**Table 2 ijms-24-07687-t002:** Baseline characteristics of SDND/FRNS patients stratified according to the occurrence of at least one NS relapse within 24 months from rituximab administration. FSGS: focal segmental glomerulosclerosis; MCD: minimal change disease; PDN: prednisone equivalents (average dose during the month preceding rituximab administration). Data are expressed as median [interquartile range] or number (percentage). *p*-values refer to the comparison between relapsers and non-relapsers.

	Overall (*n* = 18)	Relapsers (*n* = 9)	Non-Relapsers (*n* = 9)	*p*-Value
Age (years)	40.5 [27–47]	43 [26–53]	36 [31–42]	0.4529
Male sex, n (%)	10 (55%)	3 (33%)	7 (78%)	0.1534
MCD vs. FSGS	11 vs. 7	4 vs. 5	7 vs. 2	0.3348
SDNS vs. FRNS	15 vs. 3	8 vs. 1	7 vs. 2	1.0000
Pediatric onset, n (%)	8 (44%)	4 (44%)	4 (44%)	1.0000
Serum creatinine (mg/dL)	0.80 [0.75–0.94]	0.75 [0.64–0.96]	0.81 [0.58–1.29]	0.4799
Serum albumin (mg/dL)	3.7 [3.6–4.0]	3.6 [2.85–3.75]	4.0 [3.70–4.13]	0.0240
Urinary protein (g/24 h)	0.09 [0.07–0.17]	0.17 [0.05–2.47]	0.09 [0.05–0.180]	0.1709
Average PDN dose (mg/kg/day)	0.47 [0.27–0.61]	0.57 [0.36–0.80]	0.33 [0.25–0.55]	0.0851
Complete vs. Partial remission	15 vs. 3	6 vs. 3	9 vs. 0	0.2059

## Data Availability

The data presented in this study are available on request from the corresponding author.

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
