# Peer review of "Immunophenotypic Alterations in Adult Patients with Steroid-Dependent and Frequently Relapsing Nephrotic Syndrome"

_ijms, 2023, doi:10.3390/ijms24097687_

Round 1

Reviewer 1 Report

The study retrospectively analyzed immunophenotyping characteristics before in adult patients with steroid-dependent/frequently relapsing nephrotic syndrome and also compared distinctive profiles between with/without NS relapse after Rituximab use.

In general, the study is well conducted despite its retrospective and single-centre nature.

My comments as follows:

1. B/T immunophenotyping in this study is detailed. The results showed major changes including memory B subsets, Treg subsets among different groups, which have been mentioned before although in other clinical settings.  

2. Unfortunately, there is lack of follow-up data, showing dynamic changes in relapsing phase after B cell depletion and reconstitution, which could be critical in the aspects of investigating B-T interaction.

3. The sample size is relatively small, and difficult to perform subgroup analysis (for example, separating MCD and FSGS

Author Response

Dear Editors,

Dear Referees,

Thank you for your letter regarding our manuscript ijms-2290796 “IMMUNOPHENOTYPIC ALTERATIONS IN ADULT PATIENTS WITH STEROID-DEPENDENT AND FREQUENTLY-RELAPSING NEPHROTIC SYNDROME”. We appreciated the Editors’ and Referees’ time and effort in reviewing our paper.

A revised version of the manuscript is enclosed with changes marked up using “Track changes”. As requested by the Referees, we acknowledged the limitations of our study, including the small sample size and the lack of follow-up data, which would have been critical in investigating dynamic T and B cell interaction during disease relapse in the post-rituximab period. We also reported more clearly that our B/T cell immunophenotype findings have been previously described by other groups.

As requested by the Referees, we added a comparison between T and B cell subpopulations in SDNS/FRNS patients with a diagnosis of either MCD or FSGS. We did not find significant changes between these two study groups, possibly due – as reported in the revised version of our manuscript – to the small sample size. As suggested, we included a new figure reporting the experimental design and moved the captions of all the figures within the figure section after the images.    

We are grateful to the Editors and Referees for their comments that helped us to improve the manuscript, which we hope is now suitable for publication in the International Journal of Molecular Sciences.

With best regards,

Ariela Benigni, PhD

Reviewer 2 Report

This is a very well designed and described study.

The groups of patients are small and this should be clearly stressed as a limitation of the study.

It would be interesting to add a comparison between MCD and FSGS patients - are the B or T cells populations different between those 2 histologic changes?

Author Response

(The authors gave the same response as above.)

Reviewer 3 Report

Thank you very much for the opportunity to review the manuscript.

In the present translational study, the Authors investigated changes in the B and T cell phenotype of adult patients with SDND/FRNS after steroid-induced remission and whether any of these changes in immune cell subsets could discriminate between patients who developed NS relapses after steroidsparing treatment with rituximab from those who did not. They found an expansion of memory B cells and reduced memory Tregs patients with SDND/FRNS, concluding that Treg levels at baseline may help identify patients who will achieve sustained remission following rituximab infusion from those who will experience NS relapses.

The manuscript is well written and the work presented is original.The topic is original because searching new predictors of response to the treatment in nephrotic syndromes is essential.

The methodological approach is appropriate and accurate, the message is clearly driven to the reader, and the conclusions correlate to the results found. The results are clearly presented and all the conclusions are supported by the results. All the cited references are relevant to the research and well-balanced. The tables and figure correspond to the description in the text and they are well-designed and reflect important information.

Few comments that could improve the quality of the paper:

1.      A common scheme of the experiment could be helpful to the readers

2.    Captions to Figures would be moved within the Figures section after the images.

Generally, I think that this is a very worthy work. I express my gratitude to the authors for their work and my great pleasure in reading their results.

Author Response

(The authors gave the same response as above.)
